# *m*HC-lite: You Don't Need 20 Sinkhorn-Knopp Iterations

**Yongyi Yang**[*]
University of Michigan
NTT Research, Inc.
yongyi@umich.edu

**Jianyang Gao**[*]
Nanyang Technological University
jianyang.gao@ntu.edu.sg

### ABSTRACT

Hyper-Connections (HC) generalizes residual connections by introducing dynamic residual matrices that mix information across multiple residual streams, accelerating convergence in deep neural networks. However, unconstrained residual matrices can compromise training stability. To address this, DeepSeek's Manifold-Constrained Hyper-Connections (*m*HC) approximately projects these matrices onto the Birkhoff polytope via iterative Sinkhorn–Knopp (SK) normalization. We identify two limitations of this approach: (i) finite SK iterations do not guarantee exact doubly stochasticity, leaving an approximation gap that can accumulate through network depth and undermine stability; (ii) efficient SK implementation requires highly specialized CUDA kernels, raising engineering barriers and reducing portability. Motivated by the Birkhoff–von Neumann theorem, we propose ***m*HC-lite**, a simple reparameterization that explicitly constructs doubly stochastic matrices as convex combinations of permutation matrices. This approach guarantees exact doubly stochasticity by construction and can be implemented using only native matrix operations. Extensive experiments demonstrate that *m*HC-lite matches or exceeds *m*HC in performance while achieving higher training throughput with a naive implementation and eliminating the residual instabilities observed in both HC and *m*HC.

## 1 INTRODUCTION

Residual connection He et al. (2016a), which adds identity mappings between every adjacent layers, is known to be critical for stabilizing the training of deep neural networks. Latest advancements generalize a single stream of residual to multiple streams to add the flexibility of feature reuse across depth Xie et al. (2024); Zhu et al. (2024); Mak & Flanigan (2025); Bhendawade et al. (2025); Xie et al. (2025); Liu et al. (2025). Among these works, Hyper-Connections (HC) proposes to build dynamic residual matrices $\boldsymbol{H}_l^{\text{res}}$ to mix the information across residual streams, which enriches the expressive capacity of residual connections and accelerates the convergence Zhu et al. (2024).

Recently, researchers from DeepSeek observe that, as the training scales up, the unconstrained dynamic residual matrices may introduce risks of instability Xie et al. (2025). In particular, replacing the identity residual connection with a dynamic residual matrix removes the explicit guarantee of the identity property. As a result, gradients could become unstable, and exploding gradients may re-emerge when non-identity mappings are repeatedly composed across depth.

To mitigate this, Xie et al. (2025) proposes Manifold-Constrained Hyper-Connections (*m*HC), which approximately constrains the dynamic residual matrices onto the Birkhoff polytope, i.e., the set of doubly stochastic matrices. The doubly stochastic matrices have all their row and column sums being one, and thus, ensures that their spectral norm is bounded by 1 and that the set is closed under

---

[*]Equal contribution.

matrix multiplication, preventing gradient explosions in their composition in deep neural networks. In particular, *m*HC's approximate constraint is achieved via the iterative Sinkhorn–Knopp (SK) algorithm Knopp & Sinkhorn (1967), which alternately normalizes all columns and rows so that their sums equal 1.

However, *m*HC's reliance on a finite number of SK iterations creates an inherent approximation gap and raises the engineering barrier to efficient adoption. First, based on the finite number of iterations (in the *m*HC paper, 20 iterations), exact doubly stochasticity is not guaranteed. Classical results on matrix scaling establish that the SK algorithm can converge arbitrarily slowly for certain input matrices Linial et al. (1998); Knight (2008); Chakrabarty & Khanna (2021); Franklin & Lorenz (1989). Thus, under a limited number of iterations, the resulting matrices may remain noticeably away from the intended constraint, potentially undermining the stability that *m*HC targets. To make this concrete, we present a simple example adapted from Linial et al. (1998):

$$\begin{pmatrix} \frac{1}{2}, & \alpha, & \alpha \\ \frac{1}{2}, & \alpha, & \alpha \\ \alpha, & 1, & 1 \end{pmatrix} \xrightarrow{\text{SK (20 iters)}} \begin{pmatrix} 0.91, & 0.045, & 0.045 \\ 0.91, & 0.045, & 0.045 \\ 0., & 0.5, & 0.5 \end{pmatrix}$$

where the input matrix is strictly positive with $\alpha = 10^{-13}$. After 20 SK iterations, the output matrix has column sums $1.82$, $0.59$, and $0.59$, which deviates substantially from doubly stochasticity. In deep networks, such approximation errors can accumulate through depth, and repeated composition of these matrices may further deviate from the desired doubly stochasticity, which may introduce risks of stability. We include a more detailed analysis of the approximation in Section 2.

Second, *m*HC's efficiency relies on highly specialized implementations of the SK iterations, which increases engineering complexity and reduces portability across software stacks. To achieve competitive efficiency for running the SK iterations, it requires custom fused CUDA kernels to amortize repeated kernel launches in the forward pass, as described in Xie et al. (2025). Moreover, to control the memory footprint, *m*HC's implementation avoids storing per-iteration intermediate results in the SK algorithm and instead recomputes them during the backward pass. Such tightly optimized operators are less well supported by generic deep learning infrastructures. Taken together, these stability concerns and engineering barriers make *m*HC difficult to adopt as a drop-in replacement for the classical identity residual connection He et al. (2016a).

Notably, while *m*HC applies SK iterations to approximate doubly stochasticity, the *m*HC paper itself Xie et al. (2025) highlights a critical fact: the Birkhoff polytope is the convex hull of the set of permutation matrices, which is known as the Birkhoff–von Neumann theorem Birkhoff (1946); von Neumann (1953). Motivated by it, we propose ***m*HC-lite**, which parameterizes doubly stochastic matrices with a convex combination of permutation matrices, thereby bypassing SK iterations entirely. The parameterization allows us to represent any doubly stochastic matrix by an unconstrained weight. This re-parameterization yields two benefits: (i) it guarantees exact doubly stochasticity by construction, eliminating approximation errors; (ii) it can be

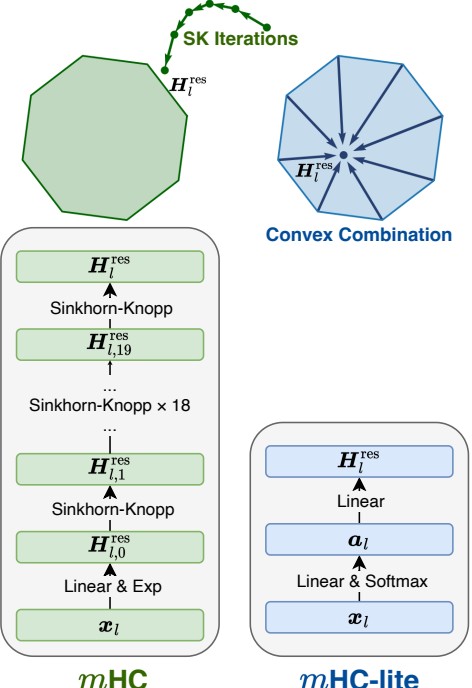

Figure 1: **Residual matrix construction in *m*HC vs. *m*HC-lite.** The method *m*HC relies on repeated Sinkhorn–Knopp iterations to approximate doubly stochastic matrices, whereas *m*HC-lite directly computes the matrix via a convex combination of permutation matrices, achieving exact doubly stochasticity.

guarantees exact doubly stochasticity by construction, eliminating approximation errors; (ii) it can be

efficiently implemented via native matrix multiplications, removing the reliance on highly specialized kernels for iterations.

We conduct extensive experiments to validate the effectiveness of *m*HC-lite. Our results highlight three key advantages. First, *m*HC-lite matches (and sometimes exceeds) the performance gains of *m*HC, demonstrating that it is a competitive alternative. Second, unlike *m*HC, *m*HC-lite maintains training throughput even with a naive, unoptimized implementation, highlighting its practicality in standard training stacks. Third, we find that *m*HC can still exhibit instability in practice (though less severe than HC), whereas *m*HC-lite eliminates this issue entirely.

In summary, our contributions are as follows:

1. We propose *m*HC-lite, a simple reparameterization of *m*HC that explicitly constructs doubly stochastic residual matrices, eliminating the requirement of SK iterations, closing the approximation gap entirely, and enabling simple and fast implementation based solely on native matrix operations.

2. We provide both theoretical and empirical evidence that finite SK iterations in *m*HC can leave a non-negligible approximation gap to the doubly stochastic constraint, showing that stability issues persist in *m*HC despite the manifold constraint.

3. Through extensive experiments, we show that *m*HC-lite matches or surpasses *m*HC in downstream performance while achieving higher training throughput and removing the instabilities of the residual matrices observed in *m*HC and HC.

**Organization.** Due to space limit, we only outline our main method in the main paper, while deferring the main experiment results and discussions to appendix. Specifically, Section 2 provides an in-depth investigation of the remaining stability concerns of *m*HC under finite SK iterations and introduces our proposed *m*HC-lite algorithm. Section A reviews the background on residual connection designs, highlighting the instability issue of HC and the manifold-constrained remedy adopted by *m*HC. Section B presents experimental results that validate our claims. Finally, Section C discusses limitations and future directions.

## 2  METHODOLOGY

As discussed in Section 1, *m*HC's reliance on a finite number of SK iterations raises concerns regarding portability and stability. From a system perspective, achieving competitive efficiency for SK iterations typically relies on specialized, fused CUDA kernels, making this component difficult to serve as a drop-in replacement for standard residual connections across different frameworks. Beyond portability, a more fundamental issue lies in the stability of the residual matrices. In particular, finite-step approximation can lead to non-negligible deviations from exact doubly stochasticity, which may accumulate across depth and undermine the stability that *m*HC aims to achieve. We analyze this stability issue in detail in Section 2.1. These observations together motivate a re-parameterization in Section 2.2, which ensures exact doubly stochasticity by construction and avoids heavy customization of CUDA kernels.

### 2.1  ANALYSIS OF THE STABILITY

In *m*HC, a fixed number of SK iterations (e.g., 20 iterations in *m*HC) does not guarantee a high-quality approximation when the convergence is slow. Classical studies on matrix scaling show that SK is not uniformly fast in general Linial et al. (1998); Knight (2008); Chakrabarty & Khanna (2021). For general nonnegative matrices, the SK algorithm only comes with a worst-case iteration bound as follows: to obtain an approximation of doubly stochasticity whose $\ell_1$-error [1] is at most $\epsilon$, it may require up to $O\left(\frac{n^2 \log(n/\nu)}{\epsilon^2}\right)$ iterations, where the relative range $\nu$ is defined by

$$\nu := \frac{\min\limits_{i,j:\, x_{i,j}>0} x_{i,j}}{\max\limits_{i,j} x_{i,j}}, \tag{1}$$

---

[1]This bound follows from Corollary 2 in Chakrabarty & Khanna (2021). Here, the $\ell_1$-error indicates the summation of the errors of all the column/row sums, i.e., $\ell_1\text{-error}(\boldsymbol{X}) := \|\boldsymbol{X}\mathbf{1}_n - \mathbf{1}_n\|_{\ell_1} + \|\boldsymbol{X}^\top\mathbf{1}_n - \mathbf{1}_n\|_{\ell_1}$.

where $x_{i,j}$ is the $(i,j)$-th entry of $\boldsymbol{X}$. Even for strictly positive matrices, convergence remains sensitive to $1/\nu$ and can be extremely slow when $1/\nu$ is large Linial et al. (1998) (see the example in Section 1).

This issue is practically relevant in $m$HC. As shown in Equation (5), the SK input is obtained by exponentiating an affine function of the features, which can yield ill-conditioned matrices with very large relative range. In our measurements (Figure 2), approximately $27.9\%$ of SK inputs satisfy $1/\nu \geq 10^{13}$. Under such inputs, a fixed SK budget may fail to produce a near-doubly-stochastic matrix. Figure 3 shows that the column sum of a single residual matrix in $m$HC may deviate from 1 by up to $100\%$. More importantly, these per-layer deviations can accumulate through depth: Figure 3 shows that the column sums of $\prod_l \boldsymbol{H}_l^{\mathrm{res}}$ may deviate from 1 by up to $220\%$ in a 24-layer network, implying the risks of instability when models further scale up. In practice, a latest model constructs a 1,000-layer network for self-supervised reinforcement learning Wang et al. (2025) based on the classical identity residual connection He et al. (2016a). This empirical trend indicates the importance of stable residual matrices with theoretical guarantees.

## 2.2 RE-PARAMETERIZATION AND $m$HC-LITE

Our methodology is based on the Birkhoff-von Neumann Theorem Birkhoff (1946); von Neumann (1953), which is also highlighted by the original $m$HC paper Xie et al. (2025). To keep the paper self-contained, we restate the theorem as follows.

**Theorem 2.1** (The Birkhoff-von Neumann theorem). *For any $\boldsymbol{X} \in \mathcal{B}_n$, there exists a weight $\boldsymbol{a} = (a_1, ..., a_{n!}) \in \mathbb{R}^{1 \times n!}$, where $a_k \geq 0, \forall k \in [n!]$ and $\|\boldsymbol{a}\|_{\ell_1} = 1$, such that $\boldsymbol{X} = \sum_{k=1}^{n!} a_k \boldsymbol{P}_k$, where $\{\boldsymbol{P}_k\}_{k=1}^{n!}$ is the sequence of all $n \times n$ permutation matrices.*

Based on the Birkhoff-von Neumann theorem, we directly represent doubly stochastic matrices as convex combinations of permutation matrices. This parameterization guarantees that the matrix is precisely doubly stochastic. Furthermore, by eliminating iterative approximations, the parameterization removes their computational overhead in both training and inferencing, avoiding the heavy reliance of highly specialized infrastructures.

In $m$HC-lite, to control for confounding factors, we keep the structure of $m$HC unchanged, except for $\boldsymbol{H}_l^{\mathrm{res}}$. Let $\boldsymbol{x}_l \in \mathbb{R}^{n \times C}$ denote the input feature in the $l$-th layer and $\hat{\boldsymbol{x}}_l \in \mathbb{R}^{1 \times nC}$ denote the flatten input feature. Then we build mappings $\boldsymbol{H}_l^{\mathrm{res}}, \boldsymbol{H}_l^{\mathrm{pre}}$ and $\boldsymbol{H}_l^{\mathrm{post}}$ dynamically based on $\boldsymbol{x}_l$ as follows.

$$\hat{\boldsymbol{x}}_l' = \mathrm{RMSNorm}(\hat{\boldsymbol{x}}_l)$$

$$\boldsymbol{H}_l^{\mathrm{pre}} = \mathrm{sigmoid}\left(\alpha_l^{\mathrm{pre}} \hat{\boldsymbol{x}}_l' \boldsymbol{W}_l^{\mathrm{pre}} + \boldsymbol{b}_l^{\mathrm{pre}}\right)$$

$$\boldsymbol{H}_l^{\mathrm{post}} = 2 \cdot \mathrm{sigmoid}\left(\alpha_l^{\mathrm{post}} \hat{\boldsymbol{x}}_l' \boldsymbol{W}_l^{\mathrm{post}} + \boldsymbol{b}_l^{\mathrm{post}}\right)$$

$$\boldsymbol{a}_l = \mathrm{softmax}\left(\alpha_l^{\mathrm{res}} \hat{\boldsymbol{x}}_l' \boldsymbol{W}_l^{\mathrm{res}} + \boldsymbol{b}_l^{\mathrm{res}}\right) \tag{2}$$

$$\boldsymbol{H}_l^{\mathrm{res}} = \sum_{k=1}^{n!} a_{l,k} \boldsymbol{P}_k \tag{3}$$

where $\boldsymbol{W}_l^{\mathrm{pre}}, \boldsymbol{W}_l^{\mathrm{post}} \in \mathbb{R}^{nC \times n}$ and $\boldsymbol{W}_l^{\mathrm{res}} \in \mathbb{R}^{nC \times n!}$ are learnable weight matrices in the $l$-th layer. Here $\boldsymbol{b}_l^{\mathrm{pre}}, \boldsymbol{b}_l^{\mathrm{post}} \in \mathbb{R}^{1 \times n}$ and $\boldsymbol{b}_l^{\mathrm{res}} \in \mathbb{R}^{1 \times n!}$ are learnable bias. The terms $\alpha_l^{\mathrm{pre}}, \alpha_l^{\mathrm{post}}$ and $\alpha_l^{\mathrm{res}}$ are learnable scalars. The $\mathrm{RMSNorm}(\cdot)$ refers to the RMSNorm Zhang & Sennrich (2019).

In the implementation, we first compute a dynamic weight vector $\boldsymbol{a}_l = (a_{l,1}, \ldots, a_{l,n!}) \in \mathbb{R}^{n!}$ via a linear layer with softmax activations. Recall that $n$ denotes the number of residual streams, which is $n = 4$ in HC and $m$HC Zhu et al. (2024); Xie et al. (2025), so $n! = 24$ is a small constant. To produce $\boldsymbol{H}_l^{\mathrm{res}}$, Equation 3 is implemented via a matrix multiplication between $\boldsymbol{a}_l^{\mathrm{res}}$ and a constant 0/1 matrix in $\mathbb{R}^{n! \times n^2}$, which is reshaped from the concatenation of all permutation matrices.

Like HC and $m$HC Xie et al. (2024); Zhu et al. (2024), the additional FLOPs introduced by the residual connection are typically negligible compared to those of the main transformation $f(\cdot; \mathcal{W}_l)$. For instance, in Transformer architectures Vaswani et al. (2017), $f(\cdot; \mathcal{W}_l)$ corresponds to the attention and MLP operator, which dominates the computation. Our key advantage in the computation, instead, is engineering-oriented: the construction can be implemented entirely with standard operators, avoiding reliance on specialized kernels for repeated iterations, and is thus more generally portable across frameworks.

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

# A  BACKGROUND

The residual connection paradigm, originally introduced by ResNet (He et al., 2016a), has been serving as the fundamental backbone of modern deep learning. It builds an identity mapping path that mitigates the vanishing gradient problem and enables the training of extremely deep networks (He et al., 2016b). This design was subsequently adopted by the Transformer architecture (Vaswani et al., 2017) and has proven essential for the scalability of large language models (LLMs), such as GPT-3 (Brown et al., 2020) and Llama (Touvron et al., 2023).

Despite its widespread success, the standard residual connection has inherent limitations. The single-stream design restricts information flow to a single pathway, potentially limiting the representational capacity of very deep networks (Huang et al., 2017). Moreover, the fixed identity mapping, while stabilizing training, offers no adaptability to the varying computational demands across different layers or input contexts (Srivastava et al., 2015). These observations have motivated recent research into more flexible and expressive connection mechanisms that go beyond the simple identity shortcut while preserving training stability (Xie et al., 2024; Zhu et al., 2024; Mak & Flanigan, 2025; Bhendawade et al., 2025; Xie et al., 2025; Liu et al., 2025).

**Hyper-Connections (HC).**  Hyper-Connections (HC) generalizes residual connections by expanding a single residual stream into multiple streams and introducing dynamic connections among these streams Zhu et al. (2024). This generalized residual connection enriches the model's connectivity and has been reported to accelerate convergence with little additional computation Zhu et al. (2024). Let $\boldsymbol{x}_l \in \mathbb{R}^{n \times C}$ denote the input feature of the $l$-th layer, where $n$ is the number of residual streams and $C$ is the dimensionality. The architecture is formulated as follows.

$$\boldsymbol{x}_{l+1} = \boldsymbol{H}_l^{\mathrm{res}} \boldsymbol{x}_l + \boldsymbol{H}_l^{\mathrm{post}} f(\boldsymbol{H}_l^{\mathrm{pre}} \boldsymbol{x}_l; \mathcal{W}_l) \tag{4}$$

where the residual matrix $\boldsymbol{H}_l^{\mathrm{res}} \in \mathbb{R}^{n \times n}$ is dynamically determined by learnable parameters and $\boldsymbol{x}_l$, and is used to mix the residual streams. The terms $\boldsymbol{H}_l^{\mathrm{pre}}, \boldsymbol{H}_l^{\mathrm{post}} \in \mathbb{R}^{1 \times n}$ are decided by learnable parameters and $\boldsymbol{x}_l$, and is used to aggregate the input and expand the output respectively. The term $f(\cdot; \mathcal{W}_l)$ represents a learnable function parameterized by weights $\mathcal{W}_l$. For the detailed computation of $\boldsymbol{H}_l^{\mathrm{res}}$ and $\boldsymbol{H}_l^{\mathrm{pre}}, \boldsymbol{H}_l^{\mathrm{post}}$ in HC, we refer readers to the original paper Zhu et al. (2024).

**Manifold-Constrained Hyper-Connections (*m*HC).**  Manifold-Constrained Hyper-Connections modifies the computation of $\boldsymbol{H}_l^{\mathrm{pre}}, \boldsymbol{H}_l^{\mathrm{post}}$ and $\boldsymbol{H}_l^{\mathrm{res}}$, particularly, attempting to constrain $\boldsymbol{H}_l^{\mathrm{res}}$ on the Birkhoff polytope $\mathcal{B}_n$, i.e., the set of doubly stochastic matrices, whose definition is as follows.

$$\mathcal{B}_n = \left\{ \boldsymbol{X} \in \mathbb{R}^{n \times n} \mid \boldsymbol{X}^\top \mathbf{1}_n = \boldsymbol{X} \mathbf{1}_n = \mathbf{1}_n, \ \boldsymbol{X} \geq 0 \right\}$$

where $\mathbf{1}_n$ denotes the all-ones vector and $\boldsymbol{X} \geq 0$ is entrywise. The doubly stochastic matrices exhibit identity-like stability because their spectral norms are bounded by 1 and the set is closed under matrix multiplication: repeated composition of doubly stochastic matrices is still doubly stochastic. Let $\boldsymbol{x}_l \in \mathbb{R}^{n \times C}$ denote the input feature in the $l$-th layer and $\hat{\boldsymbol{x}}_l \in \mathbb{R}^{1 \times nC}$ denote the flatten input feature. The computation of *m*HC is detailed as follows.

$$\hat{\boldsymbol{x}}_l' = \mathrm{RMSNorm}(\hat{\boldsymbol{x}}_l)$$
$$\boldsymbol{H}_l^{\mathrm{pre}} = \mathrm{sigmoid}\left(\alpha_l^{\mathrm{pre}} \hat{\boldsymbol{x}}_l' \boldsymbol{W}_l^{\mathrm{pre}} + \boldsymbol{b}_l^{\mathrm{pre}}\right)$$
$$\boldsymbol{H}_l^{\mathrm{post}} = 2 \cdot \mathrm{sigmoid}\left(\alpha_l^{\mathrm{post}} \hat{\boldsymbol{x}}_l' \boldsymbol{W}_l^{\mathrm{post}} + \boldsymbol{b}_l^{\mathrm{post}}\right)$$
$$\boldsymbol{H}_l^{\mathrm{res}} = \mathrm{SK}\left(\exp\left(\mathrm{mat}\left(\alpha_l^{\mathrm{res}} \hat{\boldsymbol{x}}_l' \boldsymbol{W}_l^{\mathrm{res}} + \boldsymbol{b}_l^{\mathrm{res}}\right)\right)\right) \tag{5}$$

where $\boldsymbol{W}_l^{\mathrm{pre}}, \boldsymbol{W}_l^{\mathrm{post}} \in \mathbb{R}^{nC \times n}$ and $\boldsymbol{W}_l^{\mathrm{res}} \in \mathbb{R}^{nC \times n^2}$ are learnable weight matrices in the $l$-th layer. The terms $\boldsymbol{b}_l^{\mathrm{pre}}, \boldsymbol{b}_l^{\mathrm{post}} \in \mathbb{R}^{1 \times n}$ and $\boldsymbol{b}_l^{\mathrm{res}} \in \mathbb{R}^{1 \times n^2}$ are learnable biases. The terms $\alpha_l^{\mathrm{pre}}, \alpha_l^{\mathrm{post}}$ and $\alpha_l^{\mathrm{res}}$ are learnable scalars. The function $\mathrm{mat}(\cdot)$ reshapes a matrix from $\mathbb{R}^{1 \times n^2}$ to $\mathbb{R}^{n \times n}$. The $\mathrm{RMSNorm}(\cdot)$ refers to the RMSNorm Zhang & Sennrich (2019). The $\exp(\cdot)$ function is entrywise. The $\mathrm{SK}(\cdot)$ iteration alternately rescales all columns and rows so that their sums equal 1. In the setup of *m*HC, the SK iteration is repeated 20 times.

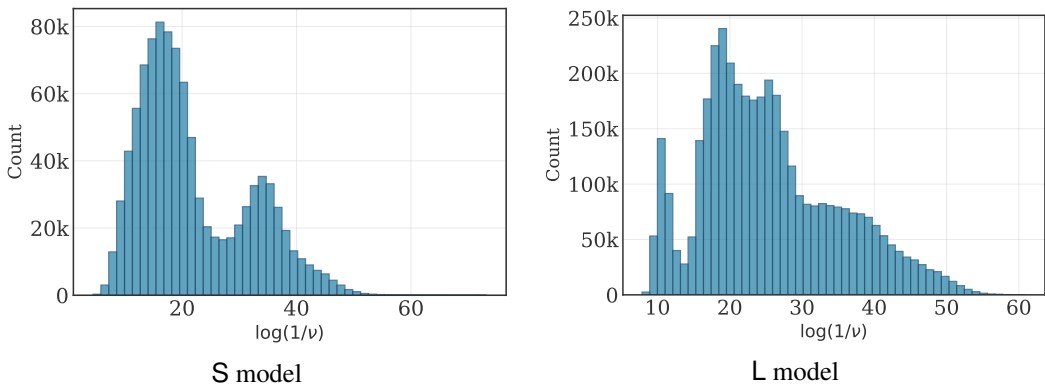

Figure 2: **Distribution of** $\log(1/\nu)$. Distribution of the relative range $\log(1/\nu)$ (defined in Equation (1)) for $m$HC before applying SK. Large values (e.g., $\log(1/\nu) > 30$) suggest that 20 SK iterations may not converge well to a doubly stochastic matrix.

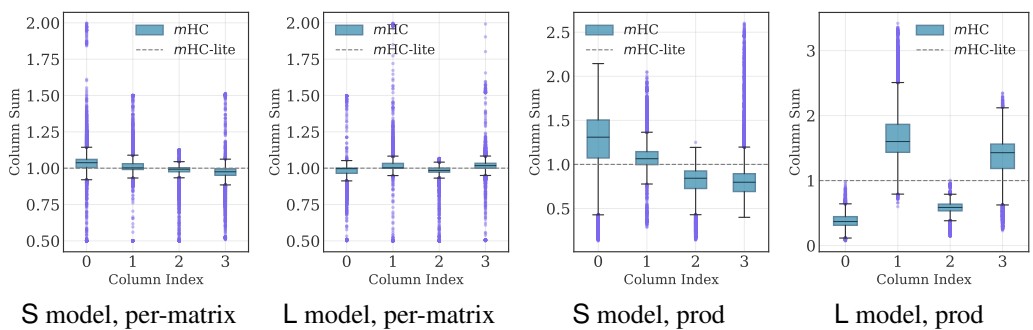

Figure 3: **Column sums of $H^{\text{res}}$.** We compute column sums for token-level $H^{\text{res}}$ matrices and summarize their distribution with standard boxplots (points indicate outliers). **per-matrix**: statistics for individual $H^{\text{res}}$ matrices. **prod**: statistics for the layer-wise product of $H^{\text{res}}$ across all layers.

## B EXPERIMENTS

To evaluate the effectiveness of $m$HC-lite, we implement $m$HC-lite in language models by replacing the original residual connections, and assess its impact on both training efficiency and model performance across various scales and datasets. Specifically, we adopt the nanoGPT framework nanoGPT (2022) and adopt three model scales: S (6 layers, $\sim$45M parameters), M (12 layers, $\sim$0.12B parameters), and L (24 layers, $\sim$0.36B parameters). For training data, we use `OpenWebText` and `FineWeb-Edu`. Following the implementation in Xie et al. (2025), throughout this paper $n$ is set to 4. Due to computational constraints, we use a relatively small number of training iterations (10,000

| Dataset | OpenWebText | | | | | | FineWeb-Edu | | | | | |
|---|---|---|---|---|---|---|---|---|---|---|---|---|
| Model Scale | S | | M | | L | | S | | M | | L | |
| | Train | Val | Train | Val | Train | Val | Train | Val | Train | Val | Train | Val |
| Residual | 3.566 | 3.562 | 3.343 | 3.336 | 3.237 | 3.242 | 3.526 | 3.536 | 3.316 | 3.321 | 3.238 | 3.240 |
| HC | 3.475 | 3.471 | 3.272 | 3.264 | 3.244 | 3.248 | 3.463 | **3.473** | 3.266 | 3.273 | 3.241 | 3.244 |
| $m$HC | 3.474 | 3.469 | 3.267 | 3.259 | **3.191** | **3.198** | 3.462 | **3.473** | **3.237** | **3.243** | 3.200 | 3.204 |
| $m$HC-lite | **3.471** | **3.467** | **3.261** | **3.255** | 3.194 | **3.198** | 3.468 | 3.477 | 3.243 | 3.249 | **3.181** | **3.185** |

Table 1: **Loss of trained models.** We report training and validation loss at the end of training. To mitigate stochastic fluctuations, training loss is computed as a moving average over the last 200 iterations.

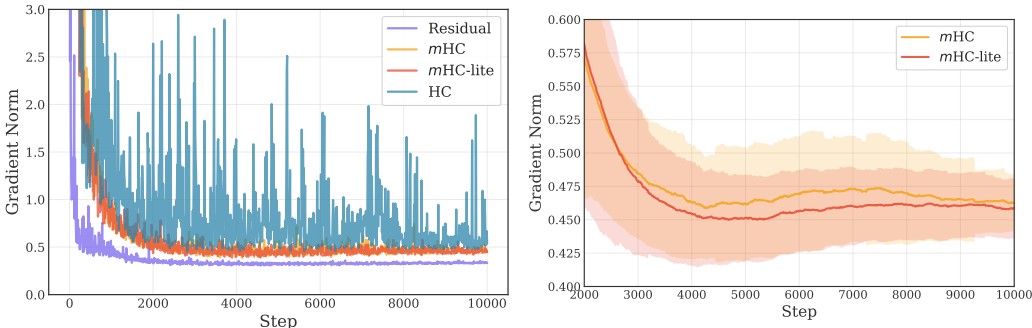

Figure 4: **Gradient-norm dynamics during training.** We compare the evolution of gradient norms over the course of training. **Left:** overall trajectories, showing that both *m*HC and *m*HC-lite exhibit substantially smaller gradient norms (and improved stability) than HC. **Right:** a zoomed-in view of *m*HC and *m*HC-lite; curves are smoothed using a 200-step moving average, and the shaded region indicates the standard deviation within the same window. From the zoomed-in view, it is clear that *m*HC-lite yields a smaller mean gradient norm and reduced fluctuations compared to *m*HC. Results are obtained with the L model on the `FineWeb-Edu` dataset.

steps, approximately 1.3B tokens in total). Further details of the hyperparameters are provided in Section D.

**Initialization.** We initialize the parameters in the HC/*m*HC/*m*HC-lite blocks so that, at initialization, each block reduces to an ordinary residual connection. Concretely, in all variants, $\boldsymbol{W}_l^{\mathrm{pre}}$, $\boldsymbol{W}_l^{\mathrm{post}}$, and $\boldsymbol{W}_l^{\mathrm{res}}$ are initialized to zero, while $\alpha_l^{\mathrm{pre}}$, $\alpha_l^{\mathrm{post}}$, and $\alpha_l^{\mathrm{res}}$ are initialized to $0.01$. The bias vectors $\boldsymbol{b}_l^{\mathrm{pre}}$ and $\boldsymbol{b}_l^{\mathrm{post}}$ are set to $-1$ in all entries except for a single entry set to $1$. For *m*HC, $\boldsymbol{b}_l^{\mathrm{res}}$ is set to $-8$ for all entries except the diagonal, which is set to $0$, so that after exponentiation it closely approximates the identity matrix. For *m*HC-lite, $\boldsymbol{b}_l^{\mathrm{res}}$ is set to $-8$ for all entries except the entry corresponding to the identity matrix, which is set to $0$, so that after the $\mathrm{softmax}$ operation the weights concentrate on the identity matrix.

### B.1 PERFORMANCE AND TRAINING STABILITY

To verify whether *m*HC-lite achieves improvements in model loss comparable to those of *m*HC, we compare the final training and validation losses of models with different residual connection components in Table 1. The results clearly demonstrate that *m*HC-lite achieves performance on par with *m*HC or even slightly better across all datasets and model scales.

Furthermore, Figure 4 presents the gradient norm curves for a specific configuration (the L model trained on `FineWeb-Edu`). The results indicate that *m*HC-lite exhibits the same stabilizing effect on training as *m*HC. Moreover, a closer examination of the curves (Figure 4 right) reveals that the gradient norm of *m*HC-lite is slightly lower than that of *m*HC, further confirming its effectiveness in stabilizing training dynamics.

### B.2 EFFICIENCY

We compare the computational efficiency of *m*HC-lite to HC by measuring the average training throughput (number of tokens per second). Results are reported in Figure 5. Unless otherwise noted, all methods are implemented by us in PyTorch under the same training setup.

We have also included the *m*HC results in Figure 5. It is important to note that Xie et al. (2025) accelerates *m*HC using a specialized kernel, which is not publicly available at the time of writing. Therefore, the *m*HC throughput reported in Figure 5 is based on our PyTorch re-implementation and may underestimate the performance achievable with custom kernels.

Even with this caveat, the authors of *m*HC claimed that with their optimized *m*HC implementation, *m*HC still incurs a $6.7\%$ overhead relative to HC Xie et al. (2025), whereas *m*HC-lite achieves higher

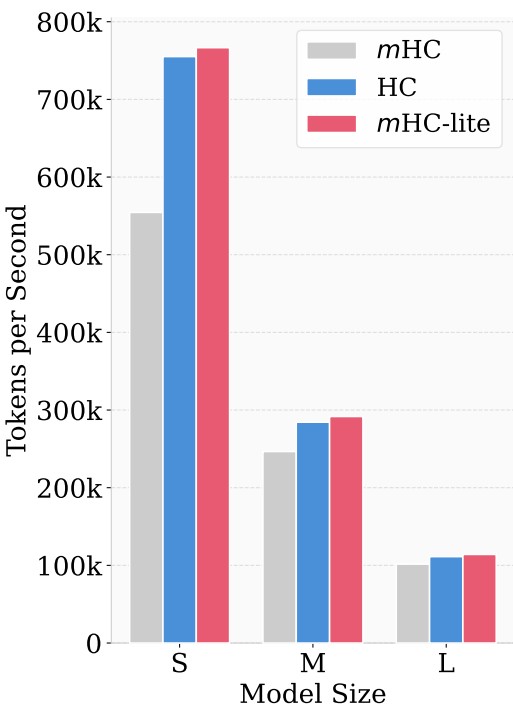

Figure 5: **Token throughput during training.** We report training throughput in tokens/s, computed as the number of tokens per batch divided by the wall-clock time of each optimizer update and averaged over the entire training run. All experiments are run on a single node with $8\times$ NVIDIA A100 80GB (SXM4) GPUs. Notice that the *m*HC result is based on our PyTorch re-implementation and may underestimate the throughput of the specialized-kernel implementation in Xie et al. (2025), which is reported to incur only a $6.7\%$ overhead relative to HC.

throughput than HC even *without any system-level optimization*. This result suggests that *m*HC-lite is highly implementation-friendly, making it easy to integrate into existing training code and practical systems.

### B.3 Stability Analysis

In this section, we address the following question: *Are the $\boldsymbol{H}_l^{res}$ matrices in mHC really as stable as claimed in Xie et al. (2025)?* To answer this, we follow the methodology in Section 5.4 of Xie et al. (2025) and assess how close $\boldsymbol{H}_l^{\text{res}}$ is to being doubly stochastic. However, rather than analyzing *token-averaged* matrices as in Xie et al. (2025), we collect matrices at each token and compute statistics over the resulting population. We argue that this procedure more faithfully reflects the behavior of $\boldsymbol{H}_l^{\text{res}}$, since averaging across tokens can hide potential instability. Concretely, for the experiments in this section, we first take the trained model and then run it on the first 64 sequences of the training set (each of length 1024). At every layer and every token, we record $\boldsymbol{H}_l^{\text{res}}$ and other related matrices, and report statistics over all collected matrices.

We begin with the relative range $1/\nu$ (defined in Equation (1)). The theoretical analysis for the SK algorithm suggests that convergence can be poor when $\log(1/\nu)$ is significantly larger than the number of SK iterations. In Figure 2, we report the distribution of $1/\nu$ for *m*HC before applying SK. The left and right panels of Figure 2 present the results for a 6-layer model and 24-layer model respectively. The results show that the fixed number of iterations, 20 iterations, taken by *m*HC, is indeed a reasonable choice for balancing the convergence rate and running time, since in many cases $\log(1/\nu) \leq 20$ or is only slightly above 20. On the other hand, however, there are also a non-negligible fraction of outliers with $\log(1/\nu) > 30$, i.e., $1/\nu > 10^{13}$, a regime in which 20 SK iterations may not converge well to the Birkhoff polytope. By comparing the left and right panels, we

further find that the relative range $1/\nu$ is generally larger for deeper models. This implies that the fixed 20 SK iterations might not be generically sufficient for deeper networks.

To show this issue more explicitly, we further directly examine the distribution of column sums of $\boldsymbol{H}_l^{\mathrm{res}}$ for $m$HC ($m$HC-lite guarantees that $\boldsymbol{H}_l^{\mathrm{res}}$ is strictly doubly stochastic). As shown in Figure 3, although the median column sum for an individual $\boldsymbol{H}_l^{\mathrm{res}}$ is typically close to 1, there exist many outliers that deviate substantially from 1. Moreover, when we consider the composition $\prod_l \boldsymbol{H}_l^{\mathrm{res}}$ across layers, even the median can drift far from 1. Similarly, by comparing the composition $\prod_l \boldsymbol{H}_l^{\mathrm{res}}$ for 6-layer models and 24-layer models, we find that the deviation is more severe when a model scales up, which implies the potential risks of instability when a model further scales up.

In contrast, $m$HC-lite does not rely on iterative normalization and therefore avoids convergence-related failure. For $m$HC-lite, the perfect doubly stochasticity of $\boldsymbol{H}_l^{\mathrm{res}}$ and its composition $\prod_l \boldsymbol{H}_l^{\mathrm{res}}$ is guaranteed by construction via the Birkhoff-von Neumann theorem.

## C  DISCUSSION

In this work, we revisit $m$HC's design of residual connections from the perspective of stability and system portability. The iterative SK algorithm requires specialized kernels for efficient execution, creating an engineering barrier for generic adoption. Moreover, through both theoretical analysis and empirical evaluation, we find that due to $m$HC's reliance on a finite steps of SK iterations, its residual matrices may significantly deviate from doubly stochasticity, when the SK algorithm fails to converge, introducing potential risks of stability. To address these limitations, we propose **$m$HC-lite**, a simple, strong, and efficient alternative to $m$HC, achieved by re-parameterizing doubly stochastic matrices based on the Birkhoff–von Neumann theorem. The re-parameterization enables us to skip the SK iterations entirely, removing the approximation gap and supporting the computation with only basic operators, making our method a drop-in replacement for classical residual architectures, offering guaranteed robustness without sacrificing ease of deployment.

The design of $m$HC-lite verifies a simple but powerful principle: exactness, when attainable, is often the most efficient form of approximation. This shift from "projection" to "reparameterization" ensures the constraint holds by construction, eliminating approximation gaps (such as those induced by finitely many Sinkhorn–Knopp iterations) while enabling potentially more efficient implementations.

**On The Computational Efficiency of $m$HC-lite for Larger $n$.**  An astute reader might notice that, although $m$HC performs well when $n = 4$, its space and time complexity grow exponentially with $n$, raising potential concerns about the efficiency of this method when $n$ is larger. Here, we make two observations: 1) in the original HC paper Zhu et al. (2024), the authors conducted extensive ablation studies demonstrating that $n = 4$ is indeed a superior choice in practice; 2) even if a larger $n$ is required, we can readily reduce the computational cost by sampling a subset of permutation matrices rather than including all of them. This is equivalent to restricting the feasible region to a subset of the Birkhoff polytope. The resulting residual matrix remains guaranteed to be doubly stochastic, while the computational budget can be tuned by controlling the number of sampled permutations.

## D  HYPERPARAMETERS

Our implementation is based on nanoGPT (nanoGPT, 2022), with all parameters set to default values unless otherwise specified. All models are trained from scratch using the AdamW optimizer (Loshchilov & Hutter, 2017) with a cosine learning rate schedule and linear warmup. We use mixed-precision training with `bfloat16` and gradient clipping. All experiments are conducted on 8 NVIDIA A100 80GB GPUs using PyTorch's DistributedDataParallel (DDP) with the NCCL backend.

The shared hyperparameters used across all experiments are summarized in Table 2.

For the three model scales (S, M, and L), their scale-specific hyperparameters listed in Table 3.

| Name | Value |
|---|---|
| batch size (per GPU) | 16 |
| block size (sequence length) | 1024 |
| # of iterations | 10000 |
| # of learning rate decay iterations | 10000 |
| # of warmup iterations | 200 |
| weight decay | 0.1 |
| $\beta_1$ | 0.9 |
| $\beta_2$ | 0.95 |
| gradient clip | 1.0 |
| dropout | 0.0 |

Table 2: Shared hyperparameters.

| Name | S | M | L |
|---|---|---|---|
| # of layers | 6 | 12 | 24 |
| # of heads | 8 | 12 | 16 |
| hidden dimension | 512 | 768 | 1024 |
| learning rate | $10^{-3}$ | $6 \times 10^{-4}$ | $3 \times 10^{-4}$ |
| minimum learning rate | $10^{-4}$ | $6 \times 10^{-5}$ | $3 \times 10^{-5}$ |

Table 3: Scale-specific hyperparameters.

