# OpenReview forum: "mHC-lite: You Don't Need 20 Sinkhorn-Knopp Iterations"
_ICLR.cc/2026/Workshop/GRaM — ICLR 2026 Workshop GRaM Poster_

### Official Review · Reviewer_v15P · 2026-02-21
**A Simple and Principled Fix to mHC**

**Rating:** 9
**Confidence:** 4

**Review:**

**Summary**

This paper proposes mHC-lite, a reparameterization of DeepSeek's Manifold-Constrained Hyper-Connections (mHC) that constructs doubly stochastic residual matrices as explicit convex combinations of permutation matrices, grounded in the Birkhoff-von Neumann theorem. The method guarantees exact doubly stochasticity by construction, requires no specialized CUDA kernels, and achieves competitive or superior performance and throughput relative to both HC and mHC.

---

**Strengths**

- The central insight is elegant and intellectually sharp. The original mHC paper invokes the Birkhoff-von Neumann theorem to justify doubly stochastic matrices, yet relies on an iterative procedure that only approximates membership in the Birkhoff polytope. mHC-lite resolves this inconsistency by exploiting the theorem constructively, parameterizing residual matrices directly as convex combinations of permutation matrices with softmax-normalized weights.

- The stability analysis is rigorous and practically grounded. Roughly 27.9% of SK inputs during training satisfy 1/ν >= 10^13, a regime in which 20 SK iterations provably fail to converge. The layer-wise product analysis further shows that per-layer deviations accumulate to up to 220% in 24-layer networks, raising well-founded concerns about mHC's scalability to very deep architectures.

- For n = 4 residual streams, the residual matrix computation reduces to a softmax over 24 logits followed by a single matrix multiplication against a precomputed constant binary matrix, eliminating the reliance on custom fused CUDA kernels and making mHC-lite a genuine drop-in replacement across heterogeneous training infrastructure.

- mHC-lite achieves higher training throughput than even vanilla HC without any system-level optimization, while matching or slightly exceeding mHC in loss across two datasets and three model scales, confirming that the reparameterization preserves expressive capacity.

---

**Weaknesses**

- The largest model evaluated is approximately 360M parameters trained on roughly 1.3B tokens, far below the regimes where the claimed stability advantages would be most consequential. The paper itself cites 1,000-layer networks as a target use case, and the trend in Figure 3 suggests deviations worsen with depth. Without larger-scale experiments, the central stability claims remain insufficiently substantiated.

- The evaluation relies exclusively on language modeling loss, which may not be the most sensitive metric for detecting the kind of stability-related degradation the authors hypothesize. Downstream evaluations such as few-shot benchmarks or held-out perplexity on diverse domains could reveal more pronounced and interpretable differences between methods. Compounding this, the loss differences in Table 1 are consistently in the third or fourth decimal place with no confidence intervals or repeated runs reported. The combination of a single narrow metric, marginal observed gaps, and absent statistical reporting makes it difficult to assess the practical significance of the results. The authors argue that differences would grow with scale, which reinforces rather than mitigates the need for both larger-scale experiments and a broader evaluation protocol.

- The finding that mHC-lite outperforms vanilla HC in throughput without any system-level optimization is surprising and practically significant, yet the paper offers no mechanistic explanation. This result is difficult to interpret or generalize without an account of why the specific computational pattern of mHC-lite aligns so favorably with framework primitives.

**Pmlr Suitability:**

NA

---

### Official Review · Reviewer_Lhve · 2026-02-24
**Variant of Manifold-Constrained Hyper-Connections**

**Rating:** 8
**Confidence:** 4

**Review:**

This paper proposes a reparameterization, grounded in Birkhoff-von Neumann theorem, of DeepSeeks mHC that explicitly constructs doubly stochastic matrices and avoids the need for specialised CUDA kernels. This leads to a method which matches or sometimes exceeds the performance of mHC with an unoptimized implementation. They also find it to be more stable. Overall the method makes a lot of sense and the empirical evaluation is convinving.

**Pmlr Suitability:**

NA

---

### Meta-Review · Area_Chair_zf1x · 2026-02-26

**Decision:**

Accept

**Metareview:**

This paper proposes an improvement upon improves Manifold-constrained Hyper-Connections. Reviewers appreciate the insight, theoretical analysis, and positive experimental results. However, the models evaluated on are relatively small, evaluations are not based on downstream benchmarks, and some significant results are lacking mechanistic explanation. I recommend acceptance and encourage the authors to incorporate reviewers’ feedbacks in the final version of the paper.

**Relevance To Proceedings:**

Tiny paper — does not apply

**Relevance To Workshop:**

Yes — suitable for GRaM

---

### Decision · Program_Chairs · 2026-03-02

Accept (Poster)